# Modelling the impact of non-pharmaceutical interventions on workplace transmission of SARS-CoV-2 in the home-delivery sector

Carl A. Whitfield[1,2,3]*, Martie van Tongeren[3,4], Yang Han[1], Hua Wei[3,4], Sarah Daniels[3,4], Martyn Regan[3,4,5], David W. Denning[2,3], Arpana Verma[3,4], Lorenzo Pellis[1], Ian Hall[1,3,6], with the University of Manchester COVID-19 Modelling Group[1¶]

1 Department of Mathematics, University of Manchester, Manchester, England, 2 Division of Infection, Immunity & Respiratory Medicine, School of Biological Sciences, University of Manchester, Manchester, England, 3 Manchester Academic Health Science Centre, University of Manchester, Manchester, England, 4 Division of Population Health, Health Services Research & Primary Care, School of Health Sciences, University of Manchester, Manchester, England, 5 National COVID-19 Response Centre, UK Health Security Agency, London, England, 6 Public Health Advice, Guidance and Expertise, UK Health Security Agency, London, England

¶ Group lead Ian Hall (ian.hall@manchester.ac.uk), a list of group members at the time of submissioncan be found in the acknowledgments.
* carl.whitfield@manchester.ac.uk

**Data Availability Statement:** The minimal required dataset to reproduce the findings in this paper are available open access, as follows: All of the

## Abstract

### Objective

We aimed to use mathematical models of SARS-COV-2 to assess the potential efficacy of non-pharmaceutical interventions on transmission in the parcel delivery and logistics sector.

### Methods

We devloped a network-based model of workplace contacts based on data and consultations from companies in the parcel delivery and logistics sectors. We used these in stochastic simulations of disease transmission to predict the probability of workplace outbreaks in this settings. Individuals in the model have different viral load trajectories based on SARS-CoV-2 in-host dynamics, which couple to their infectiousness and test positive probability over time, in order to determine the impact of testing and isolation measures.

### Results

The baseline model (without any interventions) showed different workplace infection rates for staff in different job roles. Based on our assumptions of contact patterns in the parcel delivery work setting we found that when a delivery driver was the index case, on average they infect only 0.14 other employees, while for warehouse and office workers this went up to 0.65 and 2.24 respectively. In the LIDD setting this was predicted to be 1.40, 0.98, and 1.34 respectively. Nonetheless, the vast majority of simulations resulted in 0 secondary cases among customers (even without contact-free delivery). Our results showed that a combination of social distancing, office staff working from home, and fixed driver pairings

simulated data presented in the paper can be accessed at https://doi.org/10.48420/22219564 and scripts to reproduce the plots from this data at https://doi.org/10.48420/22226266. Additionally, the source code used to generate the data can be found at https://doi.org/10.5281/zenodo.7712890, with input data at https://doi.org/10.48420/22232833. Further information regarding company consultations and their subsequent analysis can be found here https://doi.org/10.3389/fpubh.2022.864506. Full transcripts of these consultations or raw data provided by these companies (which are not required to reproduce the results in this paper) can not be shared due to commercial sensitivity and to preserve the anonymity of participants, as per the data sharing agreements and participation consent forms in place with these participants.

**Funding:** This project was funded by the UK Research and Innovation (UKRI) and National Institute for Health Research (NIHR) COVID-19 Rapid Response call, Grant Ref: MC_PC_19083. MvT is the Principal Investigator of the project. CAW and SD held post-doctoral posts funded by this grant. LP is supported by the Wellcome Trust and the Royal Society (grant no. 202562/Z/16/Z). IH is supported by the National Institute for Health Research Policy Research Programme in Operational Research (OPERA, PR-R17-0916-21001). IH and LP are supported by The Alan Turing Institute for Data Science and Artificial Intelligence, EPSRC (EP/V027468/1). CAW, IH, and LP were also supported by UKRI through the JUNIPER modelling consortium (grant no. MR/V038613/1). The funders had no role in study design, data collection and analysis, decision to publish, or preparation of the manuscript.

**Competing interests:** The authors have declared that no competing interests exist.

(all interventions carried out by the companies we consulted) reduce the risk of workplace outbreaks by 3-4 times.

## Conclusion

This work suggests that, without interventions, significant transmission could have occured in these workplaces, but that these posed minimal risk to customers. We found that identifying and isolating regular close-contacts of infectious individuals (i.e. house-share, carpools, or delivery pairs) is an efficient measure for stopping workplace outbreaks. Regular testing can make these isolation measures even more effective but also increases the number of staff isolating at one time. It is therefore more efficient to use these isolation measures in addition to social distancing and contact reduction interventions, rather than instead of, as these reduce both transmission and the number of people needing to isolate at one time.

## Introduction

Demand for home-delivery services spiked globally during the COVID-19 pandemic, as people stayed at home to reduce transmission [1]. In the UK, non-essential retail shops were closed for much of 2020 and 2021, increasing the demand for online retail and home delivery. Additionally, stay-at-home orders brought new demand for large items such as furniture and white goods as many people adjusted to spending more time at home [2]. This new and displaced demand has, on the whole, been successfully absorbed and managed by the delivery and logistics sector, due in no small part to the efforts of the key workers in those sectors to keep business moving, while adapting to a changing work environment. Meanwhile, key workers in all sectors were disproportionately exposed to transmission of SARS-CoV-2 [3]. In the delivery sector, drivers and warehouse workers were also at risk, given their exposure to a large number of contacts, the likelihood of asymptomatic transmission in SARS-CoV-2, and the potential economic impact of absence due to the prevalence of flexible or zero-hours contracts in this sector. Furthermore, studies from other countries indicate that delivery drivers there could be at much greater risk [4] than the general population, and so is a sector that requires greater attention.

Mathematical models have been central to understanding transmission of SARS-CoV-2 and in predicting the impact of various interventions. As more data has become available, models have been developed for a number of specific settings, including schools, hospitals, prisons and workplaces [5–8], to take into account the nuances and unique features of each setting. In this paper we present a model of delivery sector that has been uses to assess the impact of various measures that some companies have taken, as well as measures that were under consideration. One unique feature of these settings is the high number of brief contacts that delivery drivers have with members of the public, who themselves may otherwise have very limited contacts. Another feature in the delivery of heavy or large items is the safety requirement for employees to handle and deliver goods in pairs, often requiring prolonged close contact and entry into customers' properties. Finally, there is still the poorly understood route of fomite transmission that has the potential to be important in this setting, due to the large volume of packages being handled. The model we present considers all of these aspects, and where data is unavailable or uncertain (e.g. for risk of fomite transmission), we consider a wide range of possible scenarios.

We have developed an agent-based network model with stochastic transmission. Therefore, each realisation of the simulation represents a possible chain of transmission within a workplace, and so conclusions can only be drawn from the aggregated results of many simulations. There are commonalities with several models in the literature, including the network models for COVID-19 transmission in workplaces [8]. The stochastic infection and isolation model is similar to other agent-based and branching process models [7, 9]. The model was developed based on a combination of epidemiological data and qualitative information gained from consultations with companies in the logistics sector in the UK.

With global rollout of SARS-CoV-2 vaccines, the most severe impacts of COVID-19 on public health to be curtailed and so have most of the restrictions and measures in place to reduce transmission. However, containing the spread of new variants is likely to require good surveillance testing. There has been considerable debate around the usefulness of Lateral Flow Device (LFD) antigen tests that can be self-administered and give rapid results [10–12]. Primarily, this centres around the lower sensitivity of LFD antigen tests against Polymerase Chain Reaction (PCR) testing, particularly at low viral loads [13], and the potential impact of false positives. However, recent data suggests that LFD antigen test specificity may be at least 99.9% [14], suggesting that false positives will have a negligible impact. Furthermore, culturable SARS-CoV-2 virus is only found, at most, in the first 8–10 days following symptom onset [15–17], when viral load is higher. This suggests that lower sensitivity tests may still be useful at detecting people when they are most infectious. However, the way tests are performed (e.g. self-administered vs. trained tester) can have an impact on sensitivity [18], plus the method of rollout (e.g. supervised vs. unsupervised testing) can affect the adherence to the testing policy. The model we present accounts for these various factors.

The aim of this paper is to estimate the efficacy of different workplace interventions with a model particularly tailored to the home-delivery sector. We considered several interventions and scenarios based on formal consultations with company representative from this sector. A secondary aim is to estimate the potential impact of presenteeism (working while sick) with COVID-19 symptoms. Flexible or 'gig' contracts are common in the home delivery sector, as well as the use of self-employed couriers, all of which are factors associated with increased presenteeism [19], so this is an important factor to consider.

## Materials and methods

The project was reviewed and approved by the University Research Ethics Committee at University of Manchester, Ref: 2020-9787-15953. Consent to participation was verbally obtained before the commencement of the interviews. Written informed consent for participation was not required for this study in accordance with the national legislation and the institutional requirements.

### Data collection and company consultations

We carried out recorded consultations via teleconference with representatives from six companies between July and August 2020 (Round 1), and May and June 2021 (Round 2), three of these companies were interviewed in both rounds. Companies were recruited via engagement e-mails (via University of Manchester Business Engagement Services). Companies that volunteered then elected representatives to participate in the studies. Participants' contact details were retained by the researchers for communication purposes but no other personal information was collected or stored. Each semi-structured interview lasted 60–90 mins and was based around a set of open-ended questions regarding how the pandemic had impacted on the operations of the business and what measures had been put in place to protect staff and customers.

As part of these consultations we asked questions regarding the number of staff working at typical sites and the frequency of contacts between employees and the public. Additionally, two companies provided data on staff numbers and deliveries, which are detailed in S1.1 in S1 Text. A summary report was sent to each company for comments and corrections to verify that we had interpreted their answers accurately, and the data correctly. Further details on the consultations are published in [2]. Fitted data regarding the number of deliveries per day over time from these companies is displayed in S1 Fig.

We also used data from an online contact survey aimed at delivery drivers in the UK [20], which received 170 responses (104 of which were from the workers involved in the delivery of small packages and/or large items). This survey was elective so was not statistically representative. The results of this survey are to be published elsewhere but a few results are utilised in this paper. Namely, only 5.3% reported working while having symptoms of COVID-19 or with a member of their household having a suspected or confirmed case of COVID-19. Conversely, 17.2% reported having isolated with symptoms of COVID-19 or due to a member of their household having a suspected or confirmed case of COVID-19. This suggests approximately 1 in 4 failing to isolate for one of these reasons. For this reason we consider two $p_{\mathrm{isol}}$ values (0.5 and 0.9) as 'low' and 'high' isolation rates, noting the likely caveat of reporting biases. Staff reported large numbers of daily contacts (mean 15.0) at their place of work, which, tallying with the results of consultations, we interpreted as a result of repeated interaction within a work cohort (with only rare random interactions on top). Hence our assumed cohort size for drivers of $\approx 13$.

Finally, fitted community incidence levels for March-June 2020 were used to mimic workplace ingress rates during an active pandemic, see S2 Fig.

## Workplace network model

In this section we present an overview of the model details, with further details supplied in S1.2 in S1 Text. The model we use is a stochastic agent-based network model of disease transmission. The parameters and symbols used in the following section are all described in Table 1.

The model is parameterised to represent two archetypal delivery workplaces, a Small Parcel Delivery Depot (SPDD) and a Large-items Delivery Depot (LIDD). These represent depots that ship directly to customers. The SPDD is representative of a typical depot for (inter) national couriers shipping small packages that can be handled by a single person. The LIDD case represents a depot for logistics companies that specialise in items such as furniture and white goods, and may also offer installation/assembly of the products as part of delivery. As shown in Table 1, the LIDD model has fewer staff, longer delivery times (as the deliveries tend to be more spatially separated), longer customer contact durations (because items tend to be delivered into the home and may be assembled/installed) and thus an order of magnitude fewer deliveries per day than the SPDD model.

The model considers contacts between all employees working in a home delivery depot (i.e. engaged in business-to-consumer delivery or B2C) that has a warehouse and onsite offices. The workplace is populated by 3 groups: drivers, who deliver packages from the warehouse to customers; pickers, who transport and load packages within the warehouse; and office/admin staff, who work in the same building but in shared offices. There exists a pool of $N_D$ drivers, $N_L$ pickers and $N_O$ office staff available for work each day. Workforce turnover is ignored, as it is assumed negligible over the time scales considered, however it may play a role over long time periods.

**Table 1. Model parameters for workplace contacts and transmission.** The values given are the values used unless otherwise stated for a given figure or section. The "perceived uncertainty" is simply to indicate the level of confidence we have in the parameter values—Low: based on primary data or peer-reviewed sources; Moderate: based on literature reviews, surveys, or specific consultation questions; High: assumed or extrapolated from consultation answers.

| Parameter | Description | Value | Source | Perceived uncertainty |
|---|---|---|---|---|
| $N_D, N_L, N_O$ | Total number of drivers, pickers, and office staff employed in the workplace respectively. | SPDD: {50, 25, 15} LIDD: {20, 10, 5} | Company data and consultation | Low |
| $T_D, T_L, T_O$ | Total number of driver, picker, and office staff cohorts/teams. | SPDD: {3, 2, 1} LIDD: {2, 2, 1} | Consulations and survey | Moderate |
| $n_D(t), n_L(t), n_O(t)$ | Number of drivers, pickers, and office staff working on day $t$ respectively. | Variable | Company data and consultation | Low |
| $D_P(t)$ | Total number of packages delivered on day $t$. | Variable | Company data | Low |
| $p_c$ | Probability of two individuals at work having a random F2F contact in a given day. | $2/(N_D + N_O + N_L)$ | Consultation and survey | High |
| $\rho_D$ | For contacts including drivers, $p_c$ is scaled by this factor. | 0.05 | Consultation | Moderate |
| $f_c$ | Cohort flux rate. The probability each day of a worker switching to a different cohort. | 0.01 per day | Consultation | High |
| $\beta_{F2F}$ | Infection rate for F2F contact at 1m distance while speaking (with a person with unit infectiousness) | 0.15 h$^{-1}$ | A plausible range of 0.03–0.24 was inferred from [21–23] | Moderate |
| $c_i$ | Modifier for exposure due to type of contact $i$ | 1 (inside, talking) ×0.2 (outside) ×0.2 (not talking) | [21] | Moderate |
| $\beta_{SS}$ | Infection rate via room-sharing (with a person with unit infectiousness) | 0.002 h$^{-1}$ | See S1.4 in S1 Text | High |
| $x_{ss}$ | Effective distance for room-sharing interaction | 4.3m (shared spaces), 3.6m (office) | see S1.4 in S1 Text | Moderate |
| $\beta_{FOM}$ | Package-mediated fomite infection rate (from a person with unit infectiousness) if time between handling is 0. | 0.001 per contact | Assumed | Very high |
| $\lambda$ | Half-life of virus deposited on packages. | 3 h$^{-1}$ | [24] | Low |
| $\tau_{\text{office}}$ | Time office staff spend in shared office each day. | 6 h | Consultation | Low |
| $\tau_{\text{break}}$ | Time office and picker staff spend in shared break rooms | 1 h | Consultation | Moderate |
| $H$ | Average number of employees per employee household minus 1 ($H = 0$ means no employees live together, $H = 1$ means the average household has two employees) | 0.05, 0.5 | Assumed | High |
| $C$ | Number of cars/shared commutes per households minus 1. | 0.05, 0.5 | Assumed | High |
| $J_k[V_k(t - t_k)]$ | Relative infectiousness of person $k$ with viral load $V_k$ infected at time $t_k$. | See S1.3 in S1 Text, S3 Fig, and [25] | [26] | Moderate |
| $S_k(t - t_k)$ | Relative susceptibility of person $k$ infected at time $t_k$. | $S_k(t - t_k < 0) = 1$ $S_k(t - t_k \geq 0) = 0$ | Basic SIR model assumption | Low |
| $p_{\text{symp}}$ | Probability that an individual develops symptoms relevant for self-isolation guidance. | 0.5 | [3] | Moderate (is age, variant, and guidance dependent). |
| $p_{\text{isol}}$ | Probability that an individual adheres to self-isolation guidance. | 0.5, 0.9 | [19] and Survey | Moderate. |
| $p_{\text{miss}}$ | Probability that an 'adherent' person misses a test. | 0.4 | Based on adherence rates in the UK public sector. | Moderate. |

**Employee work schedules.** The model network consists of all within-workplace contacts between employees, as well as contacts between employees due to house-sharing or carpooling, in order to simulate workplace outbreaks in detail. Thus we assume, unless they share a household, employees only make contact with other employees if they are both at work on that day. We use an idealised model for the work schedule, whereby the number of employees in work depends on the day of the week, this pattern was calculated from data from two UK logistics firms (see S1.1 in S1 Text for details). For all pickers and drivers, we randomly assign consignments (i.e. deliveries/packages) for loading and delivery (as detailed in S1.4 in S1 Text). We

**Table 2. Summary of the direct contact routes simulated and the associated transmission rate modifier, duration of contact, and contact distance.** These are the values used in all simulations in the main text unless explicitly stated otherwise. Note that this table does not include fomite transmission routes, which are simulated and are described in detail S1.4 in S1 Text.

| Contact-type | Description | Transmission modifier ($c_i$) | Duration ($\tau_i$) | Distance |
|---|---|---|---|---|
| Cohort | F2F contacts that occur within a team or cohort (see S1.5 in S1 Text for more information). | 0.4 (either indoor or "loud-talking" outdoors, 25% of time) | Drivers: $\tau_{coh}$ = 15min Pickers/Office: $\tau_{coh}$ = 1h | 1m |
| Random | F2F contacts that occur randomly in the workplace with weighted probability towards contacts between same job roles (see S1.5 in S1 Text for more information). | 0.4 (25% of time talking) | $\tau_{rand}$ = 15 min | 1m |
| Large-item handling | Time spent lifting and moving packages in pairs (see section S1.5 in S1 Text for more information). | 0.08 (outdoor, 25% of time talking) | $\tau_{han}$ = 5 min per delivery | 1m |
| Pair delivery | Contact via sharing a cabin while delivering large-items (see S1.5 in S1 Text for more information). | 0.4 (window closed) 0.08 (window open) | $\tau_{cab}$ = 10 min per delivery | 1m |
| Pair dropoff | Contact between driver pairs during dropping large-item off at customer's property (see S1.5 in S1 Text for more information). | 0.4 (25% of time talking) | $\tau_{drop}$ = 5 min per delivery | 1m |
| Customer | Contact between driver(s) and customer during item delivery | SPDD: 0.08 (outside, talking 25% of time) LIDD: 0.24 (Inside 50% of time, talking 25% of time) | SPDD: 30s per delivery LIDD: 5min per delivery | 1m |
| Room-share | Aerosol-mediated contact in poorly ventilated rooms | 0.2 (no talking) | $\tau_{off}$ = 6h $\tau_{lun}$ = 1h | ∼ 6m (see S1.4 in S1 Text based on [27–29]) |
| Car-share | Contact via carpooling to and from work. | 0.4 (25% of time talking) | 0.5h | 1m |
| House-share | Contact via shared accommodation. | 1 | 0.5h | 1m (see S1.4 in S1 Text based on [21, 30–32])) |

assume that each consignment is first handled by pickers, then subsequently by drivers, and finally by the customer. Drivers are the only group of employees that have direct contact with members of the public while on shift. For simulation efficiency, repeat interactions with customers are not considered (as contacts via this route have a very low probability of infection, so double counting of infections is very unlikely), but these contacts are simulated and infection ingress/egress through this route is included in the model.

We also consider the case where drivers and pickers work in pairs (i.e. large goods delivery), we round the number of staff required in these roles to the nearest even number, and then assign pairings randomly each day. One intervention simulated is fixed pairings; in this case, these are assigned a priori and we pick the pairs working on a given day at random from those available. A pair is unavailable if either worker in that pair is isolating, therefore this intervention is always used alongside "pair isolation", where one member of the fixed pair isolates for the same period as their partner (whether or not they develop symptoms).

**Workplace contacts and infections.** Infections are modelled to occur via three routes; face-to-face (F2F) contact with infectious individuals, indirect contact via sharing a space with infectious individuals, and fomite transmission via goods handling.

The model generates direct F2F contacts between employees through three different mechanisms, summarised in Table 2. Table 2 also lists the parameters for the different contact routes simulated. Contacts made via these routes are assumed to be dominated by face-to-face transmission.

Indirect aerosol-mediated transmission is taken to occur on a one-to-all basis. Given the well-ventilated nature of warehouses, we assume that this kind of transmission only occurs in offices, or in lunch/break rooms. Finally, fomite transmission via package handling is

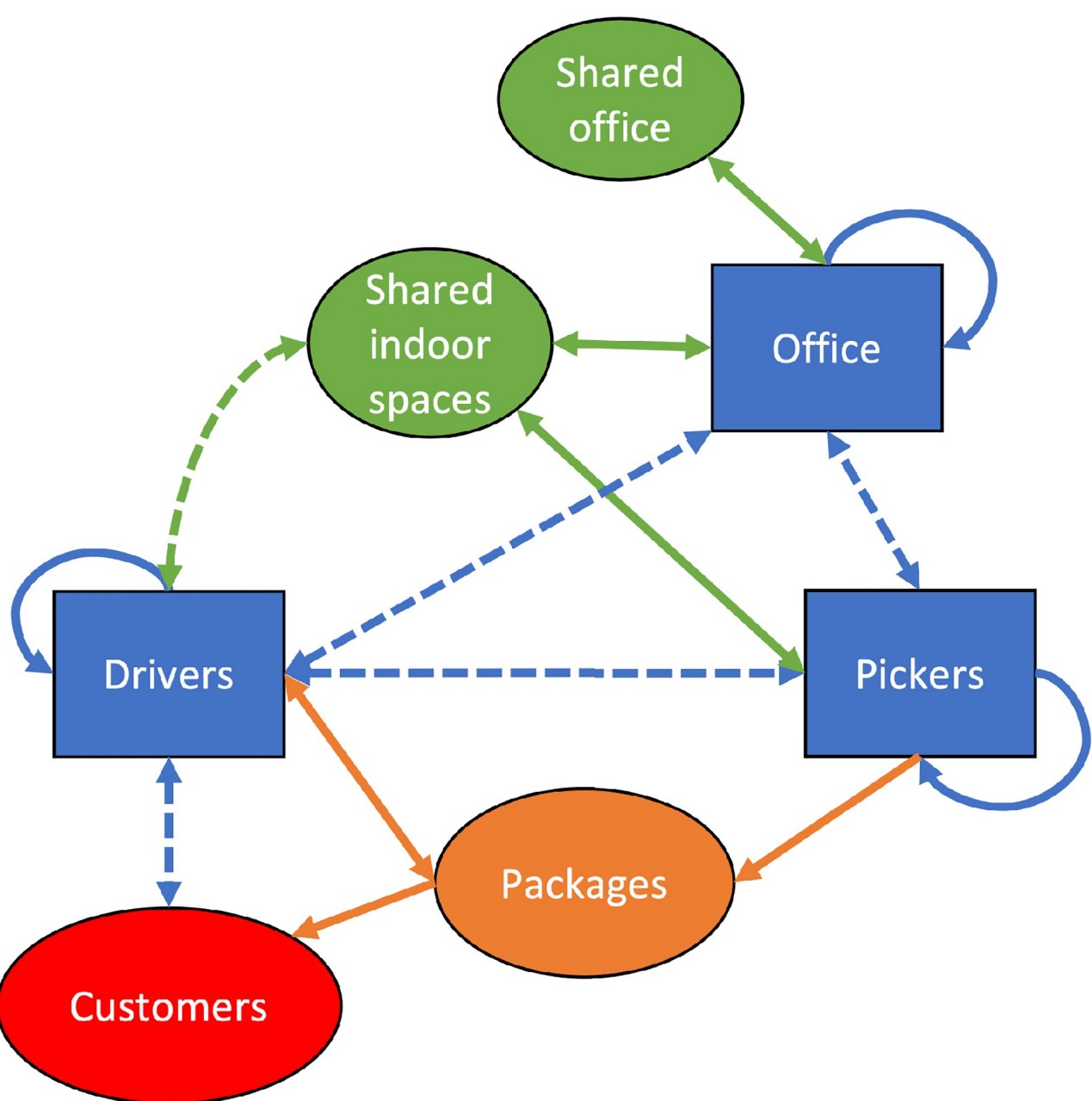

**Fig 1. Sketch of workplace staff groups and the potential transmission routes between them.** Blue lines indicate face-to-face contacts, with dashed lines indicating transmission routes with either a lower contact rate or less contact time. Orange arrows are fomite transmission routes (via packages) and green indicates aerosol transmission in shared rooms. Arrows indicate direction of transmission.

simulated as a decaying random process, such that the probability of onward transmission depends on the time between package handling events by infectious and susceptible individuals. See S1.4 in S1 Text for the justification of the various transmission parameters used.

The transmission routes between different groups are illustrated in Fig 1.

## Individual characteristics: Viral load, infectiousness, and test positivity

Viral load trajectories are generated from the individual level data in [26]. The algorithm to generate individual viral load and infectiousness profiles is described in further detail in S2 Text and in [33] is available at [34]. The method is detailed in [25] and summarised in S1.3 in S1 Text.

## Simulation algorithm

The simulations employ an individual-based network model approach with daily contact networks randomly generated using the parameterisations in Table 2. The algorithm updates contacts and infection events at discrete intervals of one day. This was chosen as the most natural option because the contact network changes from day-to-day. Additionally, the data collected to parameterise the model (including viral load data) is all defined at the scale of 1 measurement per day. However, this "synchronous" updating does introduce some error into the dynamics of the simulated epidemiology. It is known that in generic individual-based models synchronous updating can cause spurious oscillations in the dynamics compared to asynchronous methods such as a Gillespie algorithm or Markov Chain model [35]. Here a synchronous method was employed to make the model more transparent and generalisable (e.g. to non-Markovian processes), and to avoid the complexity of specifying the timings of shift and contact patterns over the course of a single day. This is similar to other recent network or IB epidemic models [5, 7, 9]. We justify this by reasoning that the error introduced is likely to be insignificant for transmission of SARS-CoV-2 as a newly infected individual is effectively non-infectious for the first day. Therefore, events where one worker is infected and then infects a co-worker within the same shift, which are missed by the synchronous update model, are vanishingly rare. Thus, there is no mechanism to trigger oscillations in this system at the timescale of the discretisation. Also, any potential effects of the artificial periodicity introduced by the simultaneous updates are obscured by the population-scale heterogeneity in infectiousness as a function of time since infections. The algorithm is outlined in detail in S2 Text.

We use the model to simulate two types of scenario:

- **Point-source outbreak:** A single index case is chosen and we assume that there are no other introductions during the simulation. All other employees are susceptible at the simulation start (i.e. zero prevalence). The simulation terminates when there are no infectious cases remaining. This type of scenario is modelled in Impact of mass testing on point-source outbreaks and in S3 Text.

- **Continuous-source outbreak:** No index cases are chosen initially and introductions occur randomly (Poisson process) based on the community incidence and prevalence in March-June 2020 (see S2 Fig). The simulation runs for a fixed time window, and the number of customer contacts and packages delivered follow the pattern of demand experienced during that period of time (see S1 Fig). This scenario is modelled in Impact of testing in the presence of household transmission.

In the point-source outbreak scenarios, in order to define a 'successful' outbreak we arbitrarily set a threshold of a final attack rate of 5%. Note that we choose this as it is a low-threshold, like the epidemiological definition of an outbreak as a single linked secondary case. However, by defining it as a percentage of workplace size this makes the results from the two different settings more comparable. Therefore, if $R - 1 > 0.05(N_D + N_L + N_O)$, where $R$ is the number of recovered individuals at the end of the simulation, then we record this simulation

as a successful outbreak. The fraction of simulations where a successful outbreak occurs is then used as an estimate of the probability of an index case resulting in an outbreak.

For continuous-source outbreaks, there is random ingress of new cases, so instead we compare the number of workplace infections (ignoring introductions) as well as the number of isolation days to measure impacts on productivity. Introductions can occur in these simulations through two routes:

- **Community ingress:** Each susceptible individual in the workplace has probability $I(t)$ of being infected outside of work, where $I(t)$ is the community incidence at time $t$.

- **Customer ingress:** For each delivery a driver makes, there is probability $P(t)$ that the customer is currently infectious, where $P(t)$ is the community prevalence at time $t$. When a susceptible driver interacts with an infectious customer, there is probability $p_{cust} = 1 - \exp(-c_i\beta_{F2F}\tau_{doorstep})$ of an infection.

This is a very simple model of case ingress and does not account for household structure, the geographical/individual variability in the wider population, or repeat deliveries to customers.

The testing strategy we model here is non-directed mass testing, i.e. all employees are tested regularly every $\tau_p$ days. A random day in the period $[1,\tau_p]$ is drawn as the first test day, and all subsequent test days follow sequentially $\tau_p$ days after the previous. Following a positive test, an individual cannot be tested again for $\tau_{pause}$ days after their positive test. Other testing strategies may be beneficial, particularly if looking to reduce the burden on employees or because of affordability, and we address some of these in the discussion.

The simulation follows an SIR-type structure, such that individuals who have previously been infected cannot be re-infected. This is a reasonable assumption over the timescales of up to 3 months that we consider here. An example visualisation of a single realisation of the simulation is shown in Fig 2. The source code for the simulations can be found at [36].

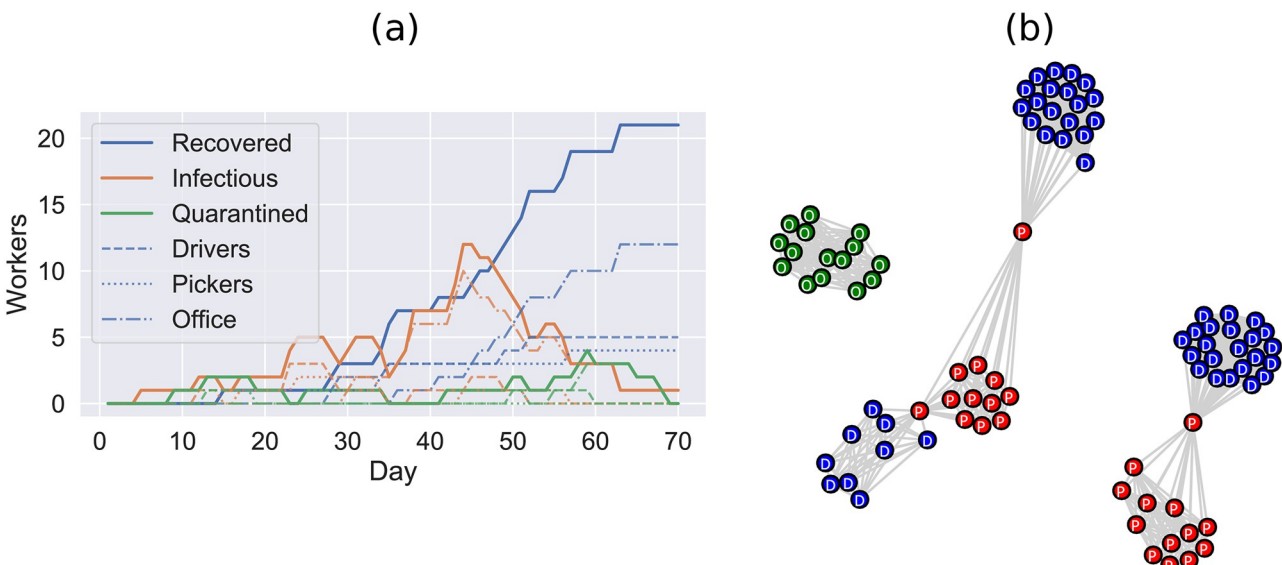

**(a)**   **(b)**

**Fig 2. Example outbreak in a SPDD workplace, where the simulation terminates when no infectious cases remain.** (a) The evolution of the number of recovered, infectious and quarantined (isolated) people in the model on each day (dashed and dotted lines indicate the same quantities for each subgroup as labelled). (b) Example network of the "cohort" contacts, each cohort has edges between all member nodes, additionally each driver cohort (blue D nodes) is supervised by a member of staff from the warehouse (red P nodes). Office staff are disconnected (green O nodes), but make contact through random interactions, break rooms, and house/car sharing arrangements.

## Results

The baseline transmission rates are summarised in S4 and S5 Figs, which show a breakdown of the mean number of staff infected by the various infection routes of the SPDD and LIDD model respectively.

S6 Fig shows the effect of the choice of work cohort size in the SPDD setting, where we predict that office size and occupancy is a more important potential factor in workplace outbreaks than transmission between drivers at the workplace, even though office workers are in the minority.

In the LIDD setting, close-contact working pairs (primarily delivery pairs, who share a vehicle for much of the day) were predicted to be very important, and keeping these pairs fixed had a significant impact on reducing workplace spread (S7(a) Fig). S7(b) Fig shows that this is also predicted to have a knock-on effect for customer infections, making them approximately as rare as in the SPDD setting.

Finally, we also present the effect of presenteeism, which in this model we define as workers with symptomatic COVID-19 attending work, which we find can have a notable effect on transmission, particularly when coupled with other measures to isolate close-contacts of symptomatic individuals (S8 and S9 Figs). S10 Fig compares the effect of presenteeism on different transmission routes and how this interacts with the fixed pairings policy.

These results (S4 Fig through to S10 Fig) are summarised in greater detail in Supplementary S3 Text.

Furthermore, S11 Fig through to S14 Fig show the sensitivity of the outbreak size to various model parameters that have significant uncertainty (namely aerosol and F2F transmission rates, fomite transmission rates, workplace size, and mixing rates between job roles). These results are summarised in Supplementary.

In this following section, we focus on the impacts of testing and the combination of different workplace interventions to analyse their potential effectiveness.

### Impact of mass testing on point-source outbreaks

Given the long incubation period of COVID-19 (compared to flu) and the significant proportion of asymptomatic cases, regular mass testing has been proposed and deployed in various settings to screen asymptomatic and pre-symptomatic cases. Figs 3 and 4 show the reduction in outbreak probability resulting from testing at different frequencies with different test types in the SPDD and LIDD work settings respectively. Overall, the results show that LFD antigen tests have a similar effect to PCR with a 2-day turnaround (given that lab turnaround targets were 24h in the UK, a 2-day turnaround was typical for much of the pandemic). Therefore, considering the relative low cost of LFDs, this suggests that they are a better option for mass testing [37]. Note this estimate for the sensitivity of LFD antigen tests is based on estimates sensitivity in phase 3b testing in [18] adjusted for the relative error induced by self vs. trained swabbing (see [25] for further details).

In each figure, two cases, representing idealised behaviours, are shown. In the first case testing is voluntary meaning 90% of people do 60% of the required tests on average (missing tests at random), while the other 10% do no tests (Figs 3(a) and 4(a)). This therefore reduces the potential benefits of testing. In the second case testing is enforced (Figs 3(b) and 4(b)) meaning that all workers test and report their results. This is the theoretical maximum effect that we could expect testing to have.

Comparing Figs 3 and 4(a), 4(b) shows that testing has a similar proportional impact in the LIDD setting. However, in the LIDD case testing has a more noticeable effect even when performed as infrequently as 14 days. With total compliance to testing, the probability of

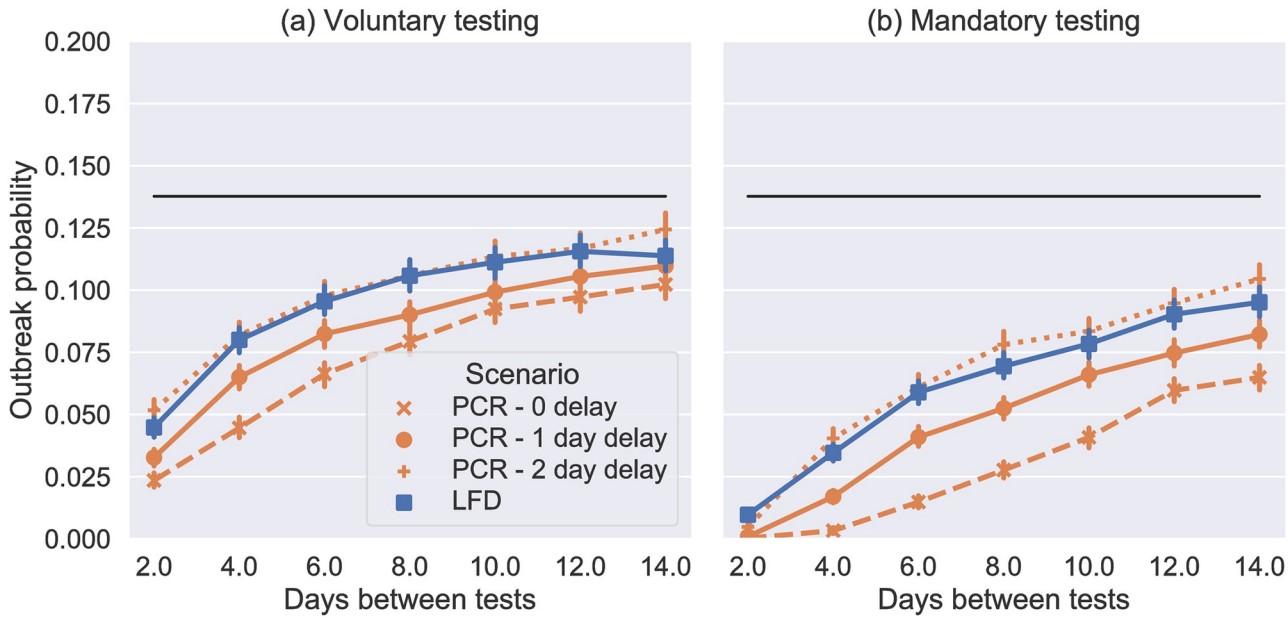

**Fig 3. Probability of an outbreak in the model SPDD setting from a single introduction (selected at random), where the black line is the mean baseline case (no testing, estimated from 10,000 simulations).** Each coloured marker shows the mean result of 10,000 simulations with the labelled testing intervention. In (a) testing is not enforced so $p_{miss} = 0.0$ and and in (b) it is so $p_{miss} = 0.0$ and all people isolate with a positive test. In both cases we use $p_{isol} = 0.9$ for symptomatic isolation.

outbreaks in both workplaces is reduced by approximately 80%, by LFD antigen tests every 4 days (which have been deployed in other sectors), see Fig 4(c). Note that this intervention is not as effective in the LIDD setting when fixed-pairing and pair isolation policies are not in place (approximately 75% reduction from a higher baseline without the fixed pairings policy, Fig 4(c)). Therefore, targeted isolation policies can improve the efficacy of testing, as well as reducing transmission rates.

To conclude, we have found that regular testing, particularly in combination with close-contact isolation, can have a very significant effect on workplace transmission. Any testing intervention needs to be weighed against potential costs, at low community prevalence the vast majority of tests are likely to be negative, and those that are positive are more likely to be false positives and so the intervention may not represent good value for money. Alternatively, at high community prevalence, testing and close-contact isolation could result in many isolations, some of which are only precautionary, which can have a huge impact on business. The latter case is not well described by the point-source outbreak considered in this section, as introductions into the workplace are more likely to occur in quick succession. Therefore in the following section we look at the impacts of a range of interventions in the case of a continuous-source outbreak.

## Impact of testing in the presence of household transmission

There are a number of confounding factors in reality that mean testing interventions may not be as effective as outlined in the previous section. One of these is the potential for household transmission between co-workers who share accommodation. In the previous sections we have considered 5% of worker households in the model to be shared, which is a significant fraction but not enough to have a large effect on transmission dynamics. It was suggested in

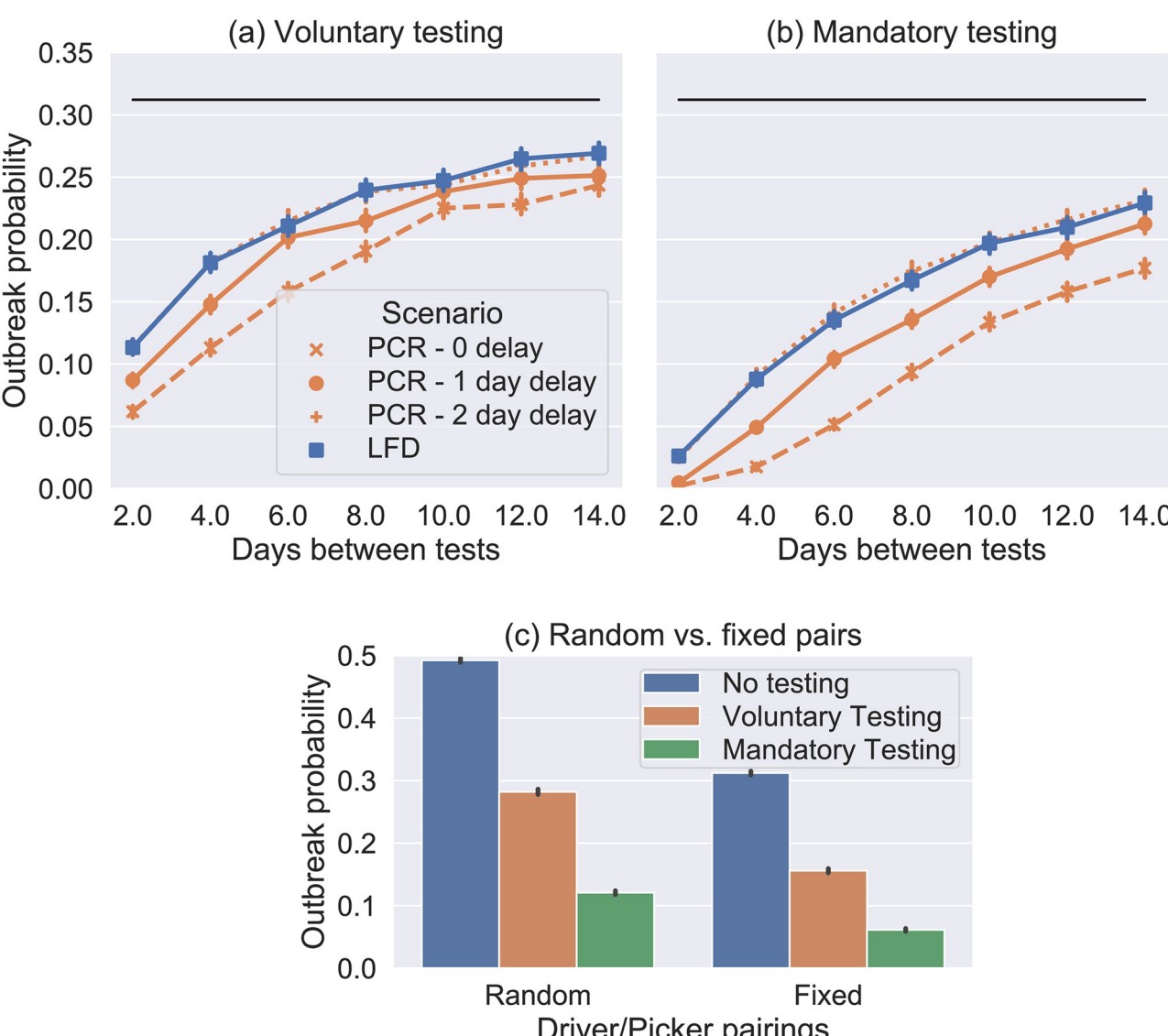

**Fig 4.** (a) and (b) show the probability of an outbreak in the model LIDD setting from a single introduction (selected at random), where the black line is the mean baseline case (no testing, estimated from 10,000 simulations). Each coloured marker shows the mean result of 10,000 simulations with the labelled testing intervention assuming that the fixed pairings and pair isolation interventions are in place for drivers and loaders. In (a) testing is not enforced so $p_{\text{miss}} = 0.4$ and and in (b) it is so $p_{\text{miss}} = 0.0$ and all people isolate with a positive test. In both cases we use $p_{\text{isol}} = 0.9$ for symptomatic isolation. (c) Bar graph comparing the outbreak of LFD antigen testing every 4 days in this setting showing both voluntary and enforced cases and both when the fixed pairings and pair isolation policies are and are not in place.

consultations that it is likely that this will vary widely by workplace location and recruitment. Therefore, in this section we test what effects changing this fraction has on these predictions.

Fig 5 shows that increasing the household sharing factor $H$ from 0.05 to 2.0 increases transmission but the relative effect of testing (regular LFD antigen testing every 3 days) remains approximately unchanged. Interestingly, a household isolation policy (i.e. the whole household isolates if one member isolates due to symptoms or a positive test) only has a minor effect for $H < 0.5$ and this is because we assume that a household transmission event between two cohabiting employees can still occur even if both are isolating (and this still contributes to the

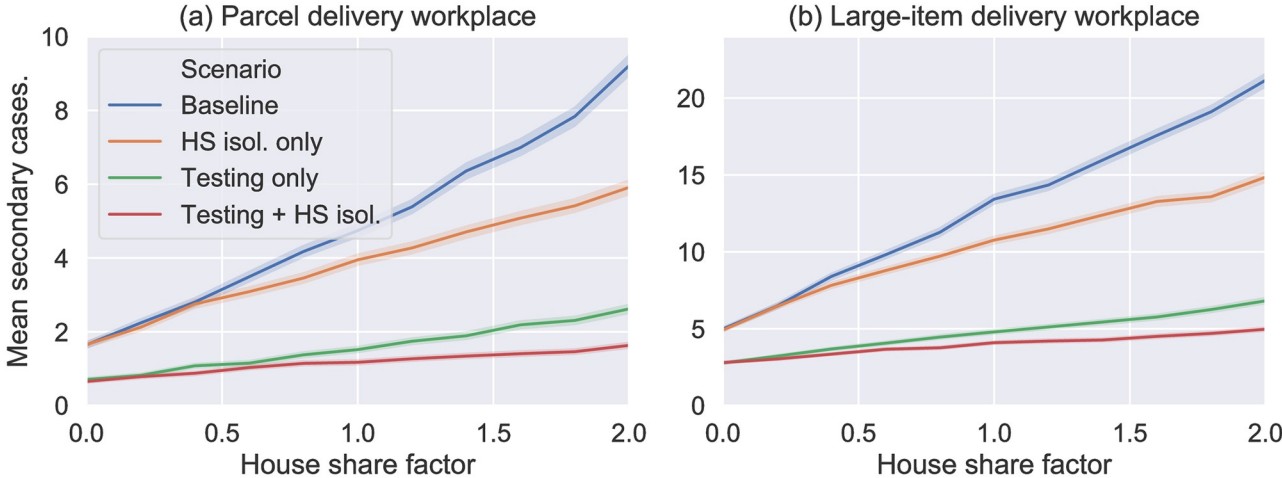

**Fig 5. Mean number of secondary cases resulting from a single random introduction plotted against the house share factor $H$.** Different colour lines show the different intervention scenarios as labelled where "Testing" means LFD antigen testing every 3 days (with default adherence rates) and "HS isol." means that the household isolation policy was implemented. (a) and (b) show the results for the two workplace types, as labelled. Each point plotted shows the mean of 10,000 simulations, with shaded error region estimated using a bootstrapping process [38].

total number of infections). In reality it is likely that this risk of household transmission may reduce during the shared isolation period if the cohabiting employees are able to remain physically separate, however it is also possible the risk will increase as they would spend more time in the shared accommodation during isolation. At very high rates of house sharing ($H > 1$ i.e. 2 or more employees in the average household), household isolation has a larger impact as this mode of transmission is prominent enough to dominate the workplace chains of transmission and household isolation can break a significant fraction of those transmission chains.

To conclude, we see that in cases where there is high rates of household-sharing (or, more generally, any contacts between employees outside of work during isolation) then this can continue to drive transmission between employees and is difficult to distinguish from workplace transmission. Nonetheless, for all values simulated, regular mass asymptomatic testing has a sizeable effect on transmission rates.

**Impact of interventions in a real-world context.** In this section we model each workplace in the context of realistic community SARS-CoV-2 incidence rates. We used incidence rates inferred from deaths and hospitalisations in the UK during the period 1st March 2020 until 31st May 2020 (see S1.2 in S1 Text). We then applied ran simulations with different interventions in place, for each scenario we added an extra intervention to the ones applied before, the interventions are:

1. Symptom isolation only: People who develop symptoms self-isolate with probability $p_{isol} = 0.5$.

2. Improved isolation: To mimic the impact of pandemic messaging, isolation probability is increased to $p_{isol} = 0.9$.

3. Distancing: All F2F interactions, except those involved in pair work, have interaction distance $x = 2$m.

4. Cohort Size Reduction: In the SPDD setting, the number of cohorts for all job types is doubled.

5. House share isolation: All employees who share a household isolate when one self-isolates.

6. Fixed-pairings: In the LIDD setting, driver and picker pairs are fixed and both self-isolate if one self-isolates.

7. Office WFH: Office staff do not enter work, they only make contact other employees if they share a household.

8. Testing: Twice weekly lateral flow testing is introduced for all employees.

9. Enforced testing: Testing becomes mandatory so no tests are missed.

10. Car share isolation: If a person travels to work with someone who self-isolates, they self-isolate.

11. Cohort isolation: If one member of the cohort isolates, all people in the cohort isolate.

In the model, introductions due to customer interactions only had a small but noticeable effect meaning that drivers were slightly more exposed than other employees (mean 0.11 introductions per driver for both work settings, averaged over all scenarios vs. 0.09 for other staff respectively). Nonetheless, over the period, around 10% of the workforce is infected purely due to the imposed prevalence and incidence.

Fig 6 shows the cumulative impact of interventions on secondary cases and isolations in the SPDD setting. The interventions are applied in approximately the sequence that was reported by companies we consulted. The intervention "Distancing" increases all "cohort" and "random" contacts to 2m interactions and has a large effect. Reducing cohort size and office staff working from home ("Office WFH") have a big impact on reducing transmission since this model predicts that outbreaks are most likely to start in this group. Interventions beyond "enforced testing" are predicted to increase isolation levels without much greater impact on transmission, particularly "cohort isolation" which likely causes a great deal of disruption despite these groups being unlikely to be infected. Note this becomes a much more viable option though if cohorts are much smaller, which is one major benefit of reducing cohort size if possible. Comparing the two graphs in Fig 6 we see that is a slightly more efficient to have contact reduction measures in place before adding isolation-based measures, as these reduce the number of workers who will need to isolate. When isolation measures are implemented alone, we see an increase in the predicted number of isolations even though the relative reduction in transmission is similar.

As shown in Impact of mass testing on point-source outbreaks, moving from voluntary to mandatory testing has a sizeable impact on transmission risk and this is reproduced here (compare "testing" to "enforced testing" in Figs 6 and 7). Interestingly, we also see it has only a small impact on the number of isolations. This is because the reduction in transmission means fewer cases, which acts to counteract the increased rate of people entering isolation. This effect is even more stark if testing is enforced in the absence of other measures (see Fig 8). In that case, the imapct of testing is significant enough to mean that the number of isolations actually reduces by switching from voluntary to mandatory testing.

The impact of interventions in the LIDD setting is very similar (see Fig 7). We see that the "fixed pairings" intervention (which includes pair isolation) has a marked effect on transmission. The extra benefit gained from testing is clearly visible too, but again isolation measures beyond this appear have little further effect.

To conclude, Figs 6 and 7 demonstrate some of the trade-offs for different intervention measures in terms of their impact on transmission and their impact on the number of isolating employees. Certain interventions act to reduce both (social distancing, Office staff WFH) but

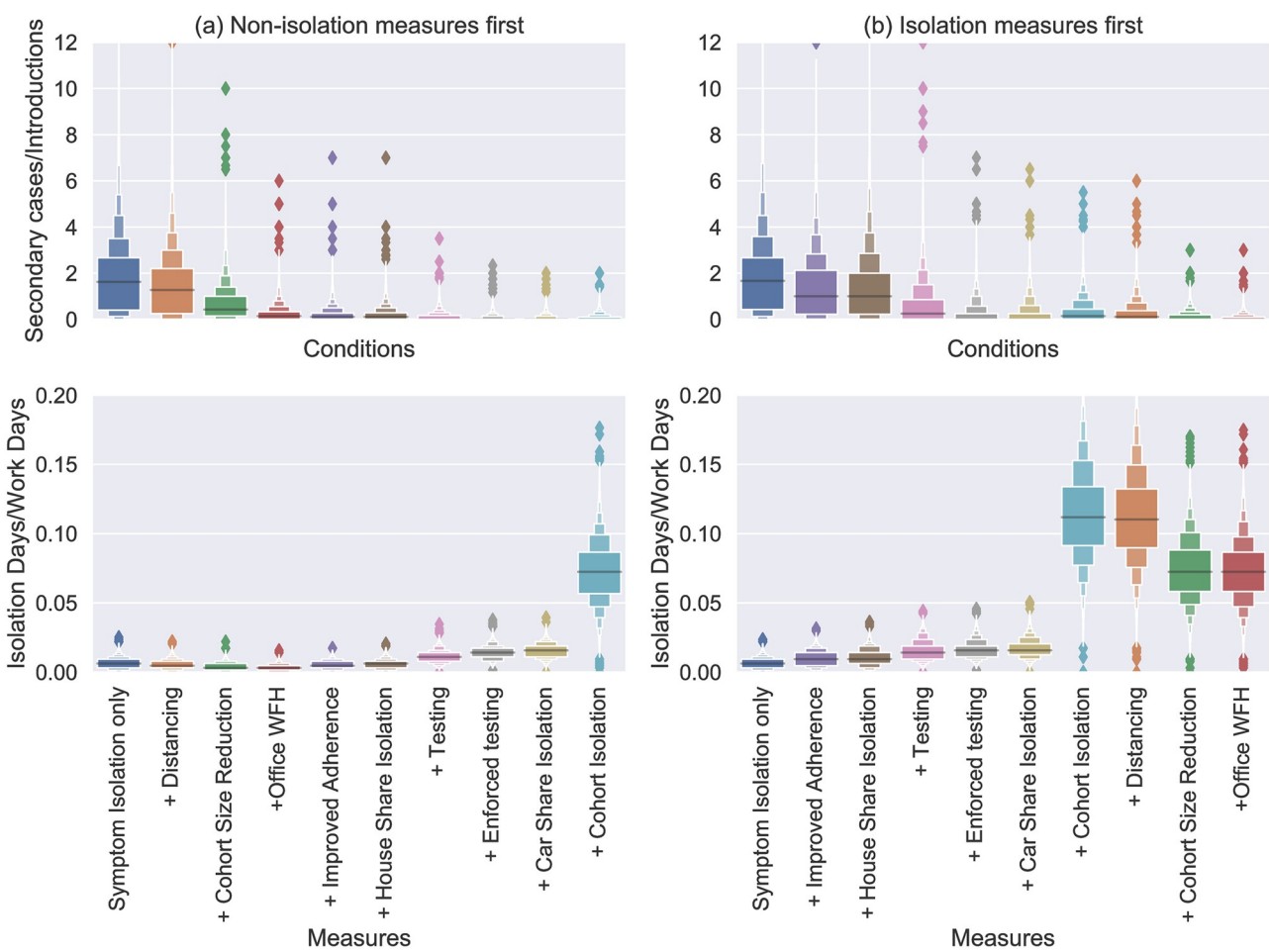

**Fig 6. Boxen plots [38] of the number of secondary cases divided by the number of introductions in a SPDD workplace over a 3 month period.** Each distribution shows all the simulations (from 10000) with more than one introduction. The labels on the *x*-axis indicate the addition of an intervention (in-tandem with all the interventions to the left). In (a) the measures restricting contacts are introduced first and in (b) the isolation-based measure.

potentially have other costs for business/feasibility issues that need to be considered. When there are employees that still need to be in close-contact (e.g. driver and picker pairs in this model) the combination of fixed pairings, pair isolation, and regular testing is highly effective for reducing transmission.

## Discussion

In this paper we have developed a stochastic model of SARS-CoV-2 spread in small/medium size workplaces. The contact patterns simulated were designed to represent warehouses/depots in the home-delivery sector, particularly those focusing on B2C delivery. To our knowledge this is the first model to consider SARS-CoV-2 transmission in this sector specifically. While the parameterisation of these models has significant uncertainty, we have been able to test the relative impact of various interventions that companies in this sector deployed to reduce SARS-CoV-2 transmission over a range of scenarios and parameter regimes.

The results predict that workplace transmission in this sector is modest, due to the bulk of the staff, drivers, working alone most of the day. Without any interventions there is predicted

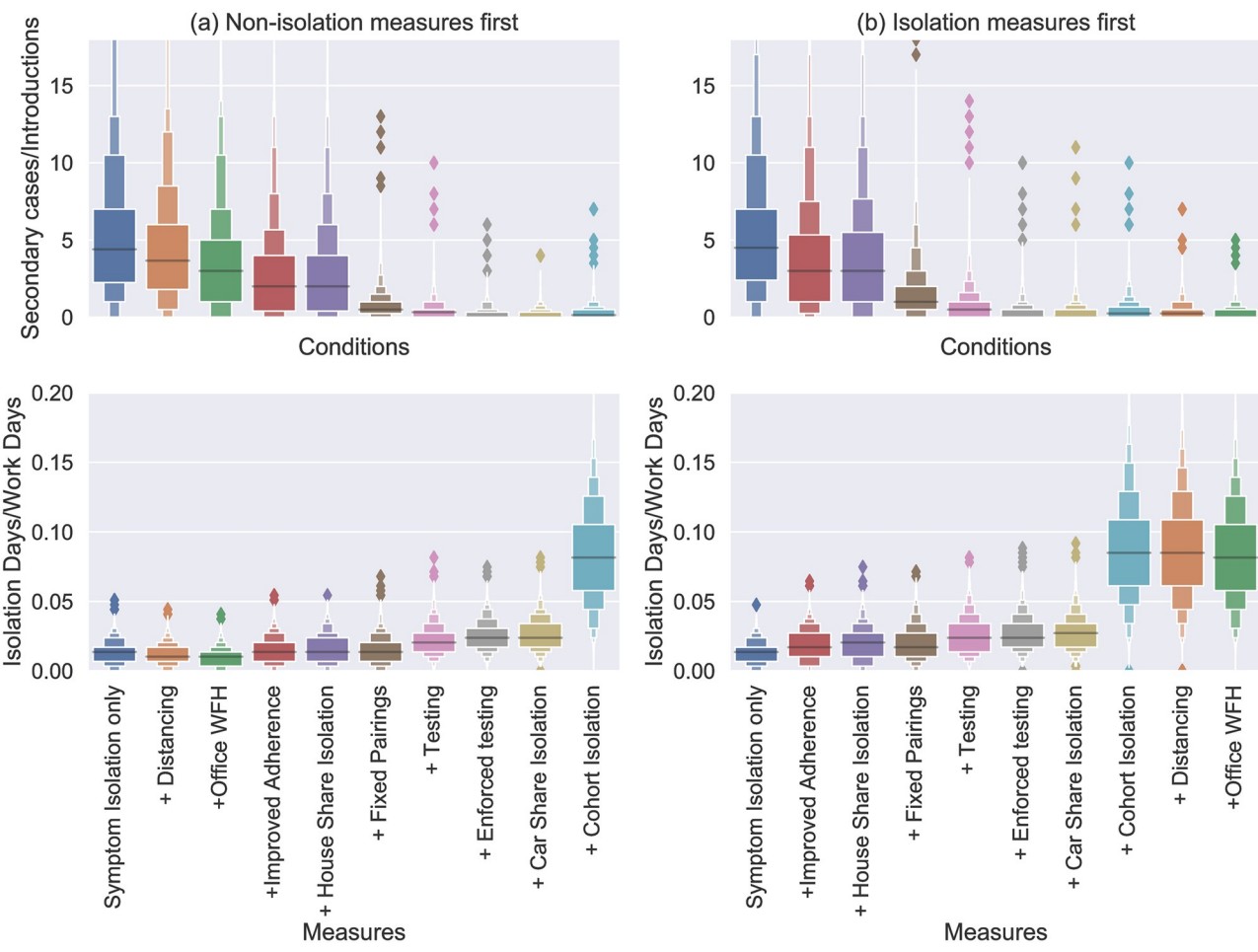

**Fig 7.** Boxen plots [38] of the rates of onward transmission (number of secondary cases divided by the number of introductions, top) and rates of isolation (number of isolation day divided by scheduled work days, bottom) in a LIDD workplace over a 3 month period. Each distribution shows all the simulations (from 10000) with more than one introduction. The labels on the *x*-axis indicate the addition of an intervention (in-tandem with all the interventions to the left). In (a) the measures restricting contacts are introduced first and in (b) the isolation-based measures.

to be a small risk to customers for an individual delivery, but in workplaces undergoing an outbreak, home-installation of items can pose a risk to customers without other interventions. The companies we consulted discontinued home-installation during in the spring of 2020, but later re-introduced it with social distancing measures. Parcel delivery companies switched to "contactless" delivery, meaning that signatures are no longer required, essentially eliminating the only route of transmission to customers. Overall, this suggests that this sector played a key role in reducing community transmission of SARS-CoV-2, as it allowed people to stay at home during periods of high-prevalence. Quantifying this impact is more difficult though as the counterfactual situation (i.e. how people would have behaved if this sector failed to keep up with increased demand or shops had remained open) is unknown.

Safeguarding the key workers in this sector was a broader challenge and companies reported implementing multiple measures based on government guidelines and their own judgement. A key result of this paper is that identifying high-risk contacts (due to e.g. shared accommodation or work tasks requiring prolonged close-contact) is very important and forms the basis of contact-tracing interventions. Workplaces have an extra advantage over contact-

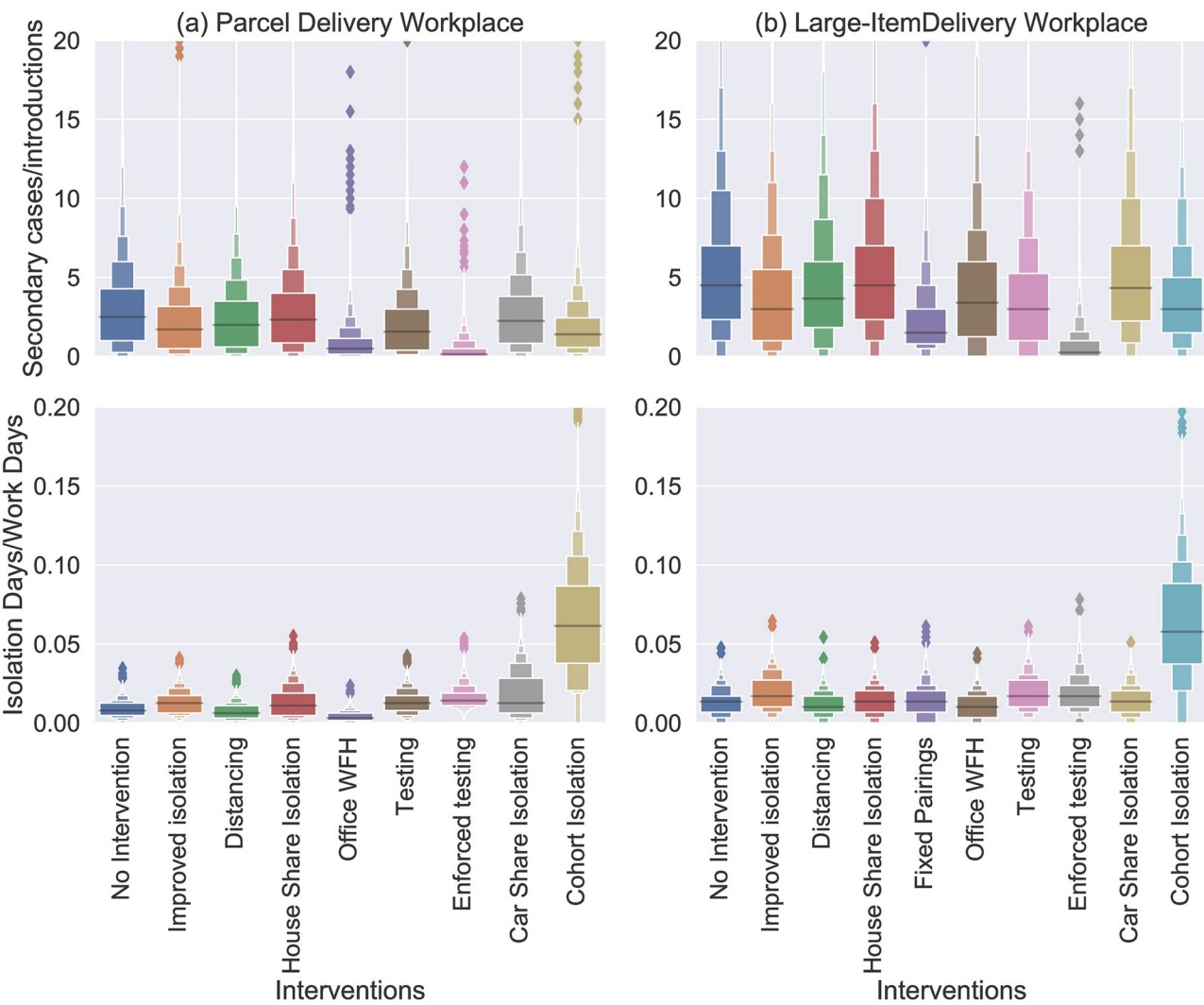

**Fig 8.** Boxen plots [38] of the rates of onward transmission (number of secondary cases divided by the number of introductions, top) and rates of isolation (number of isolation day divided by scheduled work days, bottom) over a 3 month period in (a) the SPDD work setting and (b) the LIDD work setting. Each distribution shows all the simulations (from 10000) with more than one introduction. Each distribution represents the case with application of a single intervention, as labelled on the x-axis.

tracers in that they have control and knowledge over some of the contacts that employees are required to make in their line of work. In this case, high-risk contacts can be limited by using fixed pairs for large-item delivery, reducing then number of people sharing an office, and reducing the occupancy in shared spaces. This then allows efficient isolation policies to be implemented based on knowledge of the limited number of high-risk contacts people have made (e.g. workers are given paid isolation leave if they share accommodation or work in a delivery pair with an employee who has tested positive or reported COVID-like symptoms). S7 Fig and Fig 4 show that this combination can be very effective in small workplaces.

Rates of presenteeism, which in this case we define as those who do not self-isolate when they develop symptoms, has been shown to be much less likely if fully-paid sick leave is offered [39, 40]. Therefore, in order to be effective, such isolation policies could incur considerable costs to a business as well as reducing productivity. Similarly, company-backed testing and

isolation interventions will incur further costs and mean that asymptomatic cases are detected, potentially resulting in even more isolations. Therefore, companies may be apprehensive about deploying such strategies. Here, we showed that these strategies are more efficient when close-contacts of index cases can be identified and isolated too, particularly when such contacts are necessary for the job. This can mean that chains of transmission are quickly shut down, and outbreaks are much less likely to occur. Furthermore, combining these measures with social distancing, WFH, and similar interventions that reduce transmission (e.g. masking) can reduce the number of isolations since workplace outbreaks become less likely. Combining testing measures with these can reduce risk of infection in the workplace and therefore reduce the costs of employee isolation (as fewer people will need to isolate).

The model developed has several limitations which are important for the interpretation of the results presented. First, the contact model has been developed based on a mix of quantitative (survey data, staff numbers and demand levels) and qualitative data (consultations). Novel insight was gained by speaking directly to representatives from the sector, but their position was not objective and so there may have been implicit biases in the descriptions of the nature of workplace contacts and some potential routes of infection contacts could have been missed. Furthermore, simplifying assumptions, such as all contact durations being identical for the same mode of contact, mean that this model is idealised compared to reality. This could be improved if data were available from e.g. wireless proximity sensors, as have been used in other studies to reconstruct social contact networks [41, 42], including in workplaces [43]. These provide much more high-fidelity data but when data is collected during an epidemic or while restrictions are in place, these devices can themselves affect behaviour and encourage greater distancing/policy adherence with a number of devices deployed during the pandemic actively designed to have this effect [44, 45]. Therefore, empirical contact networks in the absence and presence of restrictions are difficult to ascertain. Third, the transmission rate and the modifiers used for different types of contact are uncertain, and is based on a combination of peer-reviewed [22, 23, 29] and non-peer-reviewed literature [21]. Improvements to this transmission model from the adaption of more mechanistic modelling approaches that predict explicitly the infectious dose associated with different modes of contact [23, 29, 46–52], as well as updating with data on new variants and vaccines, will mean that this model could be applied to numerous future workplace scenarios to test the impact of different non-pharmaceutical interventions.

There are also some complicating factors we choose to ignore in this model, that may be important to consider when interpreting these results. First, we do not model severe illness, which can impact results by increasing the time away from work of individuals with COVID-19. Second, we do not model the complex relationship between interventions and behaviour. It is possible that as more interventions are introduced, adherence with other interventions wanes so the expected impact of combined interventions may not be as high as predicted. This behavioural change is difficult to predict, and so would need to be monitored by companies to gauge whether interventions are working as expected. Furthermore, even with high adherence there is no guarantee that people will use the test as intended. For example, people may be inclined to test more regularly when feeling 'run down' or 'paucisymptomatic', i.e. exhibiting very mild COVID-19 symptoms, whereas in the absence of testing they may have simply isolated from work. In this case, much of the benefit of testing can be lost [53] because asymptomatic carriers will be less likely to be detected while symptomatic carriers who would have otherwise isolated may be given a false negative and choose not to. For this reason, in some sectors, mandatory regular testing (i.e. carried out by trained swabbers at the workplace) may be the preferred option, because with the adherence rates assumed in this paper, one mandatory test per week has a similar impact to two voluntary ones (see Figs 3 and 4). To address this

shortcoming of the model, surveys of staff or test reporting rates in relevant sectors where regular testing has been deployed may inform changes. In particular, data around when and how tests were being used would be useful (as well as rates of symptomatic isolations). Survey information regarding contact frequency with other employees while off-work or in isolation would also inform the model assumptions around the effectiveness of isolation measures in reducing contacts. Finally, data from workplaces that monitor adherence to other intervention policies (such as mask-wearing) could inform the adherence rates simulated here. However with all behavioural and survey data, there is the risk of reporting bias and behavioural changes in response to observation.

One major benefit of the model presented here is that it incorporates the dynamics and variability in individual viral load, and simulates its impact on test sensitivity and infectiousness. This, means that the correlation between test-positivity and infectiousness is incorporated, meaning that impacts of these interventions can be more accurately estimated. Thus we were able to estimate not only the effect on average transmission rates, but also the frequency of rare superspreading events. This has highlighted the importance of stacking interventions that reduce transmission through different mechanisms. The source code is open access [36] and the underlying network transmission model is malleable enough to be applied to any small closed populations.

## Conclusion

This paper has shown that the multiple interventions put in place by the logistics and home delivery sector during the early stages of the pandemic are likely to have reduced the risk of workplace transmission and onward transmission into the community by safeguarding customers and staff. The availability of lateral flow tests is another valuable layer of protection that could have been added, and that this would have been most effective when combined with isolation measures that target the most high-risk contacts.

## Supporting information

**S1 Text. Sector-specific data collected and derivation of parameters.** Details the data collected and used in the simulations and derivations of the model parameters.
(PDF)

**S2 Text. Simulation algorithm.** Outlines the discretisation and simulation methods used to compute the model results.
(PDF)

**S3 Text. Baseline modelling results.** Describes the results of simulations in the baseline scenario with no interventions.
(PDF)

**S4 Text. Sensitivity analysis.** Shows the sensitivity of the model results to a number of key parameters and assumptions.
(PDF)

**S1 Fig. Parcel and large-item delivery data.** (a) Smoothed demand curves, fitted using a linear GAM, to company-wide figures for number of consignments from the parcel and logistics companies. The figures are given relative to their value at 01/03/20. (b) Weekday dependence for number of drivers and deliveries fitted using negative binomial regression. Each point shows the number of deliveries or drivers relative to the number on a Friday.
(PDF)

**S2 Fig. Rates of infection ingress.** Community incidence rates assumed for the 3-month period simulated in the continuous-source outbreak scenario.
(PDF)

**S3 Fig. Model viral load, infectiousness, and test-positive probability trajectories.** Each figure shows 50 randomly generated profiles of (a) RNA viral load ($\log_{10}$ copies/ml) and their associated (b) infectiousness (normalised units) and (c) test-positive probability. The red lines show the mean of 10,000 generated individuals at each time point (where a missing value is taken as 0).
(PDF)

**S4 Fig. Breakdown of mean secondary cases by infection route in a SPDD work setting.** Stacked bar charts of the mean number of simulated secondary infections resulting from a single index case in (a) a driver, (b) a picker, or (c) an office worker in the SPDD work setting. Each bar shows secondary infections in each group of staff broken down by transmission route, as recorded in Table 2. Note that the "shared spaces" contacts does not include contacts from sharing an office, these are counted as "cohort" interactions for office staff.
(PDF)

**S5 Fig. Breakdown of mean secondary cases by infection route in a LIDD work setting.** Stacked bar charts of the mean number of simulated secondary infections resulting from a single index case in (a) a driver, (b) a picker, or (c) an office worker in the LIDD setting. Each bar shows secondary infections in each group of staff broken down by transmission route, as recorded in Table 2. Note that the "shared spaces" contacts does not include contacts from sharing an office, these are counted as "cohort" interactions for office staff.
(PDF)

**S6 Fig. Baseline outbreak probability in a SPDD work setting.** Estimated probability of outbreak (defined as more than 3 secondary cases) resulting from a single index case plotted against the cohort flux $f_c$ in days$^{-1}$. Each marker shows the mean of 10,000 simulations, with shaded error region estimated using a bootstrapping process [38]. Point-source outbreaks where the source case was (a) a driver, (b) a picker; (c) an office worker. Each line in each figure compares simulations with different numbers of teams used for that job role, shown as the number of workers per team on average. In each figure, the job roles not shown have the default team size and $p_{\text{isol}} = 0.9$ is assumed.
(PDF)

**S7 Fig. Baseline outbreak probability in a LIDD work setting.** (a) Simulated probability of an outbreak (defined as more than 2 secondary cases). Four scenarios are shown: no intervention (staff are randomly paired each day); driver pairs travel with window open (transmission rate constant reduced to 1/5 of original value in this setting); fixed pairs (people always work with the same partner); and both of these interventions simultaneously (fixed pairs and windows open). Each bar represents 10,000 simulations, error bars indicate uncertainty in the mean, estimated via a bootstrapping method [38]. (b) Boxen plots of the number of customers infected per point-source outbreak simulation in the LIDD setting with either no or both interventions and the parcel delivery setting with default parameters.
(PDF)

**S8 Fig. Effects of presenteeism on transmission in the SPDD work setting model.** Dependence of simulated outbreak probability on the self-isolation adherence probability $p_{\text{isol}}$. The different curves show the effect of increasing the house-sharing factor $H$ as labelled.
(PDF)

**S9 Fig. Effects of presenteeism on transmission in the LIDD work setting model.** Dependence of mean number of simulated secondary cases from a single index case on the self-isolation adherence probability $p_{isol}$. The different curves show the effect of adding a fixed-pairs isolation intervention.
(PDF)

**S10 Fig. Breakdown of transmission routes for varying presenteeism in the LIDD work setting model.** Mean number of infected drivers per simulation with a single driver index case plotted against symptomatic isolation probability $p_{isol}$. The infections are broken down by those cased by close contact pair work, and all other contact routes. (a) The case with no fixed pairing intervention so pairs switch randomly each day. (b) The case with fixed pairings a pair isolation policy. Dots show the mean number of infections while shading shows 95% confidence in the mean calculated via bootstrapping methods.
(PDF)

**S11 Fig. Sensitivity to face-to-face and aerosol mediated transmission rates.** Histograms of secondary cases resulting from a single index case in the two work settings simulated for different rates of F2F and aerosol transmission. The top row shows the parcel work setting, while the bottom row is the large-item setting. For each set of simulations, the transmission rate for F2F contacts is multiplied by "F2F scale factor", and the transmission rate for aerosol contacts is multiplied by "Aerosol scale factor". Note that for the large-item workplace we assume that the fixed-pair isolation intervention is applied and in both cases $p_{isol} = 0.9$. We also assume that the index case is selected randomly.
(PDF)

**S12 Fig. Sensitivity to fomite mediated transmission rates.** The mean number of secondary cases resulting from a single index case in the two workplace types plotted for 3 values of $\beta_{FOM}$ at varying levels of demand for deliveries (x-axis). Note that for the large-item workplace we assume that the fixed-pair isolation intervention is applied and in both cases $p_{isol} = 0.9$.
(PDF)

**S13 Fig. Sensitivity to workplace size.** The mean number of secondary cases resulting from a point-source outbreak in the two workplace types plotted against workplace scale factor. Note that for the large-item workplace we assume that the fixed-pair isolation intervention is applied and in both cases $p_{isol} = 0.9$. We assume the index case is selected at random.
(PDF)

**S14 Fig. Sensitivity to mixing rates between workers in different job roles.** Histograms of the number of secondary cases resulting from a single index case in the two workplace types plotted for different scalings of $p_c(N_D + N_L + N_O)$. The top row shows the parcel delivery setting, while the bottom row is large-item setting, and each column is for the index-case labelled. Note that for the large-item workplace we assume that the fixed-pair isolation intervention is applied and in both cases $p_{isol} = 0.9$. Note also the logarithmic scale.
(PDF)

## Acknowledgments

The authors would like to thank the project's advisory group that consist of Catherine Noakes, Chris Armitage, Sheena Johnson, Jeanette Edwards, Barbara Hockey, Nina Day, Nick Gent and Thomas House, for their advice that helped refine the aims and objectives of this article.

We would also like to thank Helen Beers and Peter Baldwin from HSE for their advice on the business engagement and feedback to companies.

In addition Carl A. Whitfield, Yang Han, Lorenzo Pellis and Ian Hall (group lead), the University of Manchester COVID-19 Modelling Group includes the following authors in recognition of their equal contribution to this work: Jacob Curran-Sebastian, Rajenki Das, Elizabeth Fearon, Martyn Fyles, Thomas A. House, Hugo Lewkowicz, Christopher E. Overton, Xiaoxi Pang, Heather Riley, Francesca Scarabel, Helena B. Stage, Bindu Vekaria, Luke Webb, Feng Xu, Jingsi Xu.

## Author Contributions

**Conceptualization:** Carl A. Whitfield, Martie van Tongeren, Yang Han, Hua Wei, Martyn Regan, David W. Denning, Arpana Verma, Ian Hall.

**Data curation:** Carl A. Whitfield, Lorenzo Pellis.

**Formal analysis:** Carl A. Whitfield, Hua Wei, Sarah Daniels.

**Funding acquisition:** Martie van Tongeren, Arpana Verma, Ian Hall.

**Investigation:** Carl A. Whitfield, Hua Wei, Sarah Daniels, Ian Hall.

**Methodology:** Carl A. Whitfield, Martie van Tongeren, Yang Han, Hua Wei, Sarah Daniels, Martyn Regan, David W. Denning, Ian Hall.

**Project administration:** Hua Wei.

**Resources:** Lorenzo Pellis.

**Software:** Carl A. Whitfield.

**Supervision:** Martie van Tongeren, Yang Han, Martyn Regan, David W. Denning, Arpana Verma, Ian Hall.

**Validation:** Carl A. Whitfield.

**Visualization:** Carl A. Whitfield.

**Writing – original draft:** Carl A. Whitfield, Martie van Tongeren, Yang Han, Hua Wei, Sarah Daniels, Martyn Regan, David W. Denning, Arpana Verma, Ian Hall.

**Writing – review & editing:** Carl A. Whitfield, Martie van Tongeren, Yang Han, Hua Wei, Sarah Daniels, Martyn Regan, David W. Denning, Arpana Verma, Ian Hall.

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
