## [Decision Letter · Decision Letter 0]

24 Jan 2023

PONE-D-22-29161Modelling the impact of non-pharmaceutical interventions on workplace transmission of SARS-CoV-2 in the home-delivery sectorPLOS ONE

Dear Dr. Whitfield,

Thank you for submitting your manuscript to PLOS ONE. After careful consideration, we feel that it has merit but does not fully meet PLOS ONE’s publication criteria as it currently stands. Therefore, we invite you to submit a revised version of the manuscript that addresses the points raised during the review process.

We look forward to receiving your revised manuscript.

Kind regards,

José Ramos-Castañeda, M.Sc., Ph.D

Academic Editor

PLOS ONE

Journal Requirements:

4. One of the noted authors is a group or consortium “University of Manchester COVID-19 Modelling Group”. In addition to naming the author group, please list the individual authors and affiliations within this group in the acknowledgments section of your manuscript. Please also indicate clearly a lead author for this group along with a contact email address.

Additional Editor Comments:

Please consider the reviewers' recommendations and resubmit the manuscript.

Reviewers' comments:

Reviewer's Responses to Questions

**Comments to the Author**

1. Is the manuscript technically sound, and do the data support the conclusions?

Reviewer #1: Yes

Reviewer #2: Yes

2. Has the statistical analysis been performed appropriately and rigorously? 

Reviewer #1: Yes

Reviewer #2: Yes

3. Have the authors made all data underlying the findings in their manuscript fully available?

Reviewer #1: Yes

Reviewer #2: Yes

4. Is the manuscript presented in an intelligible fashion and written in standard English?

Reviewer #1: Yes

Reviewer #2: Yes

5. Review Comments to the Author

Reviewer #1: Abstract:

Conclusion:

Since the mathematical models developed in this study did not evaluate the transmission between workers and customers, I suggest removing this sentence “but that these posed minimal risk to customers”, or to re-write “However, these could pose minimal risk to customers”.

Introduction:

Pag 4, paragraph 2, line 7: I suggest changing: “epidemiological data and data and …” for: “epidemiological data and …”

Pag 4, paragraph 3, line 4: Since there are antibody and antigen lateral flow tests, I suggest adding “antigen”.

Methods:

Pag 10, paragraph 2: Household sharing is not a route of transmission. In this situation, the transmission occurs by the other three routes mentioned (F2F contact [droplets], indirect contact [air transmission], or fomite transmission). Household sharing is another condition such as car-share o room-share. For these reasons, I suggest removing this route from the sentence.

I suggest including the definition of parcel the delivery workplace and the large-item delivery workplace (How many workers does each workplace has?).

Results:

Pag 17, paragraph 1: Review “… xx%...”

Figure 7. I suggest changing the title. For example, “Rate of Secondary cases and isolation days in a large-item delivery workplace over a 3-month period. Also, I suggest adding a footnote such as in Figure 6.

Figure 8. I suggest changing the title. For example, “Rate of Secondary cases and isolation days by each intervention (rather than cumulatively). (a) Parcel delivery workplace. (b) large-item delivery workplace.”

Discussion:

Pag 26, paragraph 3, line 3:

Since the mathematical models developed in this study did not evaluate the transmission between workers and customers, I suggest removing this part of the sentence “the risk of community”.

Reviewer #2: The authors propose an agent-based model of SARS-COV-2 contagion in the workplace. The new model, which is parametrized using data and consultations, is used to study the effect of non-pharmaceutical interventions in the parcel delivery and logistics sector. Authors provide a detailed analysis of different interventions. Noteworthy, the model takes into account variability in the host viral load. The supplementary information provides complete details on all modeling decisions: data collection, parameter derivation, simulation algorithm, and baseline modeling. Appendix D: provides an analysis of sensitivity to assumptions and parameter choices. This is a good paper, and it should have broad applicability. However, this reviewer recommends the authors to address a few issues listed below before the manuscript is considered for publication (Please see the attached file).

6. PLOS authors have the option to publish the peer review history of their article (what does this mean?). If published, this will include your full peer review and any attached files.

Reviewer #1: **Yes: **Ruth Martínez-Vega

Reviewer #2: No

---

## [Author Response · Author response to Decision Letter 0]

27 Mar 2023

The response to reviewers has been uploaded as an attachment

---

## [Decision Letter · Decision Letter 1]

10 Apr 2023

Modelling the impact of non-pharmaceutical interventions on workplace transmission of SARS-CoV-2 in the home-delivery sector

PONE-D-22-29161R1

Dear Dr. Whitfield,

We’re pleased to inform you that your manuscript has been judged scientifically suitable for publication and will be formally accepted for publication once it meets all outstanding technical requirements.

Kind regards,

José Ramos-Castañeda, M.Sc., Ph.D

Academic Editor

PLOS ONE

Reviewers' comments:

Reviewer's Responses to Questions

**Comments to the Author**

1. If the authors have adequately addressed your comments raised in a previous round of review and you feel that this manuscript is now acceptable for publication, you may indicate that here to bypass the “Comments to the Author” section, enter your conflict of interest statement in the “Confidential to Editor” section, and submit your "Accept" recommendation.

Reviewer #1: All comments have been addressed

Reviewer #2: All comments have been addressed

2. Is the manuscript technically sound, and do the data support the conclusions?

Reviewer #1: Yes

Reviewer #2: Yes

3. Has the statistical analysis been performed appropriately and rigorously? 

Reviewer #1: Yes

Reviewer #2: Yes

4. Have the authors made all data underlying the findings in their manuscript fully available?

Reviewer #1: Yes

Reviewer #2: Yes

5. Is the manuscript presented in an intelligible fashion and written in standard English?

Reviewer #1: Yes

Reviewer #2: Yes

6. Review Comments to the Author

Reviewer #1: (No Response)

Reviewer #2: The authors have addressed all my concerns. I recommend proceeding with the manuscript publication. I believe this is a good manuscript, that might be useful to the readers of PLOS ONE.

7. PLOS authors have the option to publish the peer review history of their article (what does this mean?). If published, this will include your full peer review and any attached files.

Reviewer #1: **Yes: **Ruth Aralí Martínez-Vega

Reviewer #2: No

---

## [Editor Report · Acceptance letter]

27 Apr 2023

PONE-D-22-29161R1 

Modelling the impact of non-pharmaceutical interventions on workplace transmission of SARS-CoV-2 in the home-delivery sector 

Dear Dr. Whitfield:

I'm pleased to inform you that your manuscript has been deemed suitable for publication in PLOS ONE. Congratulations! Your manuscript is now with our production department. 

Kind regards, 

on behalf of

Dr. José Ramos-Castañeda 

Academic Editor

PLOS ONE